# Analytical ab initio hessian from a deep learning potential for transition state optimization

Eric C.-Y. Yuan [1,2,9], Anup Kumar[3,9], Xingyi Guan[1,2], Eric D. Hermes [4], Andrew S. Rosen [5,6], Judit Zádor [7], Teresa Head-Gordon [1,2,8] ✉ & Samuel M. Blau [3] ✉

Identifying transition states—saddle points on the potential energy surface connecting reactant and product minima—is central to predicting kinetic barriers and understanding chemical reaction mechanisms. In this work, we train a fully differentiable equivariant neural network potential, NewtonNet, on thousands of organic reactions and derive the analytical Hessians. By reducing the computational cost by several orders of magnitude relative to the density functional theory (DFT) ab initio source, we can afford to use the learned Hessians at every step for the saddle point optimizations. We show that the full machine learned (ML) Hessian robustly finds the transition states of 240 unseen organic reactions, even when the quality of the initial guess structures are degraded, while reducing the number of optimization steps to convergence by 2–3× compared to the quasi-Newton DFT and ML methods. All data generation, NewtonNet model, and ML transition state finding methods are available in an automated workflow.

Computational identification of transition states (TSs) on the quantum mechanical potential energy surface (PES) is central to predicting reaction barriers and understanding chemical reactivity[1,2]. The height of the barrier exponentially impacts the reaction rate coefficient via the Eyring equation, and the geometric character of the metastable state is informative about the kinetic mechanism, making TSs key to describing a broad range of chemical kinetic outcomes for enzymes, next-generation synthetic catalysts, batteries, and conformational changes of molecules and materials[3,4].

Transition states are first-order saddle points, and locating them via mathematical optimization is particularly challenging on high dimensional PESs relevant in complex molecular systems. Locating an equilibrium geometry, i.e., a local minimum on the PES, can be found using bracketing methods based on function evaluations (0th order)[5,6]

and with methods that use gradient information, such as steepest descent[7] or conjugate gradient[8] (1st order). Under the quadratic approximation, identification of the local minima can be more robustly achieved in fewer steps by 2nd order methods[9] using the Hessian matrix, whose elements $H_{ij}$ are defined as the second derivative of the energy $E$ with respect to atomic positions $R_i$ and $R_j$. However, a metastable first-order saddle point is characterized by a single negative Hessian eigenvalue, and hence 2nd order methods are indispensable to optimize for molecular TS energies and geometries[1]. The Newton-Raphson (NR) method and its variants, including restricted and augmented methods like the trust radius method (TRM) and the rational function optimization (RFO) method, select the displacement vector $\Delta R^{(k)}$ (or in internal coordinates) at step $k$ using the inverse Hessian $[H^{(k)}]^{-1}$ and the gradient $g^{(k)}$[10–12]. The geometry direct inversion of the

[1]Kenneth S. Pitzer Theory Center and Department of Chemistry, University of California, Berkeley, CA, USA. [2]Chemical Sciences Division, Lawrence Berkeley National Laboratory, Berkeley, CA, USA. [3]Energy Technologies Area, Lawrence Berkeley National Laboratory, Berkeley, CA, USA. [4]Quantum-Si, Branford, CT, USA. [5]Materials Sciences Division, Lawrence Berkeley National Laboratory, Berkeley, CA, USA. [6]Department of Materials Science and Engineering, University of California, Berkeley, CA, USA. [7]Combustion Research Facility, Sandia National Laboratories, Livermore, CA, USA. [8]Departments of Bioengineering and Chemical and Biomolecular Engineering, University of California, Berkeley, CA, USA. [9]These authors contributed equally: Eric C.-Y. Yuan, Anup Kumar. ✉e-mail: thg@berkeley.edu; smblau@lbl.gov

iterative subspaces (GDIIS) method and its variants similarly utilize Hessians and gradients to define a search space for optimization[13].

However, the evaluation of analytical Hessians for ab initio methods such as density functional theory (DFT) requires solving coupled-perturbed equations, which scale one power of system size $N$ higher than the energy or the gradient and thus can be prohibitively expensive. Consequently, almost all TS optimization approaches rely on constructing cheaper approximate Hessians using only gradient information to avoid expensive Hessian calculations, in general referred to as quasi-Newton (QN) methods[14–20]. Arguably the most widely used Hessian approximation for minimization is the Broyden–Fletcher–Goldfarb–Shanno (BFGS) method, where the Hessian is iteratively updated using a rank-2 matrix generated from the displacements and gradients. Such Hessian updates, however, are positive definite by design and thus cannot be applied to TS searches. Instead, methods such as symmetric rank-one (SR1) or Murtagh–Sargent, Powell–symmetric-Broyden (PSB), Murtagh–Sargent–Powell (MSP), Bofill, and TS-BFGS methods are developed for an indefinite approximate Hessian[21–25]. On a complex PES, an optimization step can displace the molecule from the preceding quadratic region such that the resulting updated approximate Hessian quickly diverges from the true one and thus requires expensive reconstruction[12]. Even though double- and single-ended interpolation methods such as nudged elastic band (NEB)[26,27], quadratic synchronous transit (QST)[28], and growing string method (GSM)[29] have been well-established in the field in the past few decades, a subsequent TS optimization with QN updates is often integrated into the workflow. Despite all these efforts, TS optimization still requires significant user involvement and relies on trial and error when robust Hessian information is absent. However, if the full Hessian is available at every optimization step, concerns regarding the quality of the initial Hessian and subsequent updates become much less of a problem when determining a TS.

The recent development of deep learning models for the PES provides an alternative possibility for acquiring and applying the Hessian in chemically relevant tasks[30–34]. Intuitively, the power of a fully differentiable machine learning (ML) force field does not stop at forces or gradients but also broadly applies to second (and higher order) derivative properties such as the Hessian matrix $H_{ij}$. In this case, it is possible to calculate Hessians analytically by automatic differentiation, by finite differences using gradients from the machine learning model, or by estimating Hessians using first order information as per the Davidson procedure[1]. For example, such an idea has recently been explored using Gaussian process regression, where an ML PES was locally trained on semiempirical energies, forces, and optionally Hessians and used to estimate the updated Hessians[35,36]. Yet, the high memory demand using kernel-based methods can significantly reduce the applicability on all but small systems, and the semi-empirical level of theory can be deficient for reliable chemistry.

In this work, we fine-tune an equivariant message-passing neural network (eMPNN), NewtonNet[32], on an augmented version of the Transition-1X (T1x) dataset[37], a benchmark dataset containing ~10 million configurations generated by the NEB method on ~10 thousand gas-phase organic reactions evaluated with DFT. Although the full training data is comprised of only energies and gradients of the molecular configurations, with no Hessians provided, the whole neural network is fully differentiable such that we can infer the Hessian $H_{ij}$ through back propagation. We then apply the ML Hessians to TS optimization on an independent data set of 240 organic reactions previously proposed by Hermes and co-workers[38,39], and which are outside of the training set. We have adapted the Sella code[39] to read in full ML Hessians to perform TS optimizations for these reactions and utilize the same code and optimization settings in order to compare against QN Hessian optimization with either ML or DFT.

We find that incorporating explicit Hessians from the NewtonNet ML model into TS optimization yields a 2–3 × reduction in search steps compared to approximate Hessian methods, demonstrating a remarkable efficiency improvement by ensuring higher-confidence search directions that are closer to the optimal path. The more accurate description of the Hessian also leads to improved robustness against structural perturbation such that the TS optimization is less reliant on a good initial guess. With our deep learning model, the Hessian calculation is over 1000× faster than the corresponding ab initio calculation and is consistently more robust in finding TSs than QN methods using the ML or DFT PES. The combination of greater efficiency, reduced reliance on good initial guesses, and robust TS convergence for unseen reactions opens opportunities to utilize full Hessians for TS optimizations with appropriately constructed data sets of complex reactive chemistry.

## Results

### Machine learned prediction of DFT hessians

Figure 1a shows the NewtonNet eMPNN model in which the DFT-computed molecular energy $E$ is predicted by transforming and aggregating atomic features $a_i$ that accumulate local chemical environmental information from spatial neighbors $a_j$ and interatomic distances $R_{ij}$ through message passing layers[32]. The molecular energy $E$ is then differentiated with respect to the atomic positions $R_i$ to predict atomic forces $F_i$ or gradients $g_i$, but of relevance here is that it can be auto-differentiated twice to obtain $H_{ij}$. We have demonstrated that the energies and forces can be predicted with excellent accuracy across a whole range of chemistry including small organic molecules[32] as well as for methane and hydrogen combustion, even with a limited amount of training examples[32,40].

Like all ML potentials, the quality of the learned PES and its derivative properties depends on the availability of relevant training data. Our ML model is pre-trained on the ANI-1 dataset, which contains more than 20 million off-equilibrium conformations of small organic molecules up to 8 heavy atoms and is evaluated with the $\omega$B97X density functional[41] and 6-31G* basis set[42]. Figure 1b demonstrates that the original ANI-1 dataset is mostly composed of near-equilibrium geometries[42,43] and that the reaction pathways are notably undersampled around the metastable states of the reactions[37]. As a result, the pre-trained ML model predicts the energies and forces accurately (with respect to the underlying DFT data) at the reactant and product states but fails significantly around the TS (Fig. 1c, d).

Hence, it is fortunate to have the T1x dataset[37], which is a benchmark for TS-related ML tasks, containing 9,644,740 molecular configurations generated by NEB from 10,073 organic reactions, at a level of DFT commensurate with the ANI-1 data. This data better represents the entire reaction pathway as seen in Fig. 1b and allows us to fine-tune the pre-trained model. The fine-tuned model predicts both energies and forces an order of magnitude more accurately around the TS, shown in Fig. 1c, d and S1. When using the fine-tuned model, the reaction barrier is also more smoothly interpolated between NEB images, the false identification of an energy maximum has been eliminated, and the atomic forces are correctly predicted to be negligible as expected of a first-order saddle point.

Due to the high cost of the Hessian calculation and storage, the training datasets we use do not contain ab initio Hessian reference samples. Despite the lack of such training examples, a prediction of the atomic forces from an ML model that is continuous and smooth strongly suggests the possibility of achieving Hessian predictions without explicit training on such tasks. Based on this assumption, one approach is a finite-difference Hessian estimation that can be easily realized using the gradients predicted by our model by stepping along each Cartesian axis. However, an analytical gradient of the first derivative is more cost-effective than a finite-difference method, and such a gradient can be performed as long as the neural network is at least

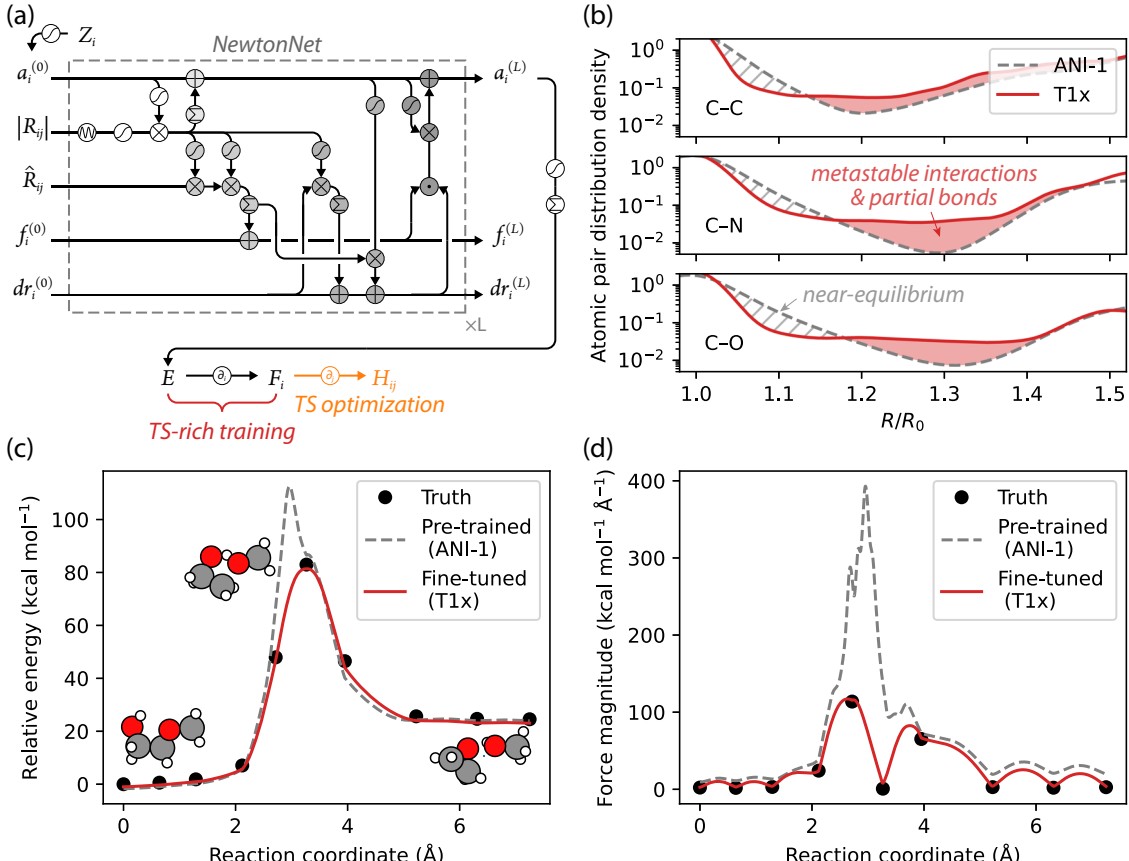

**Fig. 1 | The NewtonNet model and its performance on the ANI-1 and T1x data sets. a** The equivariant message-passing neural network designed for 3D molecular graphs with nodes $\{Z_i\}$ and edges $\{R_{ij}\}$ to predict molecular energies $E$ and atomic forces $F_i$[32]. In this work, we further differentiate the network to derive $H_{ij}$ for TS optimization tasks. **b** The distribution of atomic pairwise distances, $R$, relative to equilibrium bond distances, $R_0$, among datasets we used for training, where the T1x data set provides more data in the TS region, and further augmented with the ANI-1x data to add data corresponding to bond compression. The predicted (**c**)

potential energies and (**d**) forces along the reaction coordinate for an unseen reaction for the pre-trained and fine-tuned model. A comprehensive statistical analysis of energy and force prediction errors along the reaction coordinates for 1248 unseen test reactions is summarized in Supplementary Fig. S1 for the pre-trained and fine-tuned models. Details of the training protocols are described in Methods and Supplementary Fig. S2 and S3. TS: transition state. Source data for this figure are provided with this paper.

twice differentiable. In this regard, the NewtonNet model is designed using sigmoid linear unit (SiLU)[44] and a polynomial cutoff function[45] and is therefore $C^2$ continuous. Utilizing the automatic differentiation in our neural network, the forces and Hessians can be analytically acquired by back propagation.

Leveraging the smoothness of the deep learning PES after fine-tuning, Fig. 2 shows that reasonably accurate Hessian predictions of the reference DFT model can be acquired on molecular TSs when compared to the pre-trained model. The Hessian prediction is quantitatively accurate for both negative to positive eigenvalue regions, with eigenvalue root mean square error (RMSE) of 318 kcal mol$^{-1}$ Å$^{-2}$ and eigenvector mean cosine similarity (MCS) 0.828 for TSs in unseen test reactions, improving dramatically after fine-tuning the ML model with the T1x dataset from Fig. 2a, b. It is worth noting that the majority of the error arises from the positive eigenspace with a constant 20% underestimation of Hessian eigenvalues from DFT. This can be understood in part from Fig. 1b that the T1x dataset we use for training is biased toward weaker bonds and greater anharmonicity, such that lower apparent bond strength and force constants will be observed and modeled. We therefore augmented the T1x dataset by selecting 1,232,469 molecules from the ANI-1x dataset[43] that share the same chemical formula with the T1x dataset but that exhibit compressed chemical bonds, which further improves accuracy of the Hessian predictions to 267 kcal mol$^{-1}$ Å$^{-2}$ eigenvalue RMSE and 0.839 eigenvector

MCS in Fig. 2c. We observe that this improvement is not solely attributed to increased data volume. It effectively mitigates the underprediction of positive eigenvalues by establishing a more balanced dataset. Most importantly, the predicted Hessians by our fine-tuned model using the augmented T1x data have very accurate leftmost eigenvalue and eigenvector, which are the most critical ingredients in TS optimization and iterative Hessian diagonalization[12,46].

**Transition state optimization using machine learned Hessians**
The fine-tuned NewtonNet model for predicting TS properties is subsequently employed in practical TS optimization scenarios involving new reactions independent of the augmented ANI-1x/T1x training and test data. These include hydrogen migration reactions, endo- and exo-cyclization, generalized Korcek step 2 reactions, retro-ene reactions, and reverse 1,2 and 1,3 insertions (see source data); given the training data these involve only closed-shell molecules. We focus on TS optimization for these unseen reactions in order to compare a traditional QN method that approximates Hessians using gradient information from DFT calculations or ML predictions versus a full explicit ML Hessian used at every step.

We interfaced our fine-tuned NewtonNet model with Sella, a state-of-the-art open-source TS geometry optimizer[39]. In Sella, the interconversion between the Cartesian coordinates and the redundant

internal coordinates is automatically handled, and the Hessian is iteratively diagonalized for the leftmost eigenvector[12] used in the geodesic saddle point optimization[47]. In order to start the TS optimization for the 240 Sella benchmark reactions, we generated initial guesses with KinBot using reaction templates[38], where each template also defines the intended reactant and product end states for a given reaction. We employed restricted step partitioned rational function optimization (RS-PRFO)[48–50] for the TS optimizations, with dynamically adjusted step sizes determined by evaluating the confidence of each step (see Methods for details). After TS optimization, we follow the intrinsic reaction coordinate (IRC) from the optimized TS structure to find the minimum energy path that connects the reactant and product; the robustness of the TS optimization methods is quantified by comparing the intended reactions and the predicted reactions. The complete list of found transition states of the 240 predicted benchmark reactions is summarized in Supplementary Table S1.

Figure 3 shows how optimization efficiency is dramatically improved by providing full explicit Hessians at every optimization step, which is now affordable relative to DFT as illustrated in Supplementary Fig. S9. In Fig. 3a we find that the number of steps required to converge to a TS can be reduced by 2× of that required by the QN approach using the ML (or DFT; see Supplementary Fig. S10). The trend is notably non-linear, and full-Hessian optimization is even more advantageous for challenging tasks that require larger numbers of optimization steps. If the iterative diagonalization steps for initial Hessian construction and Hessian reconstruction when the QN approximation breaks down are included, a reduction of close to 3× of the required steps is observed when considering these gradient calls. We also observed that in some most difficult cases for TS optimization with QN Hessians that take > 80 steps, the optimization steps taken by full Hessians are even fewer than those with QN steps < 80, which initially seemed counter-intuitive. This behavior not only

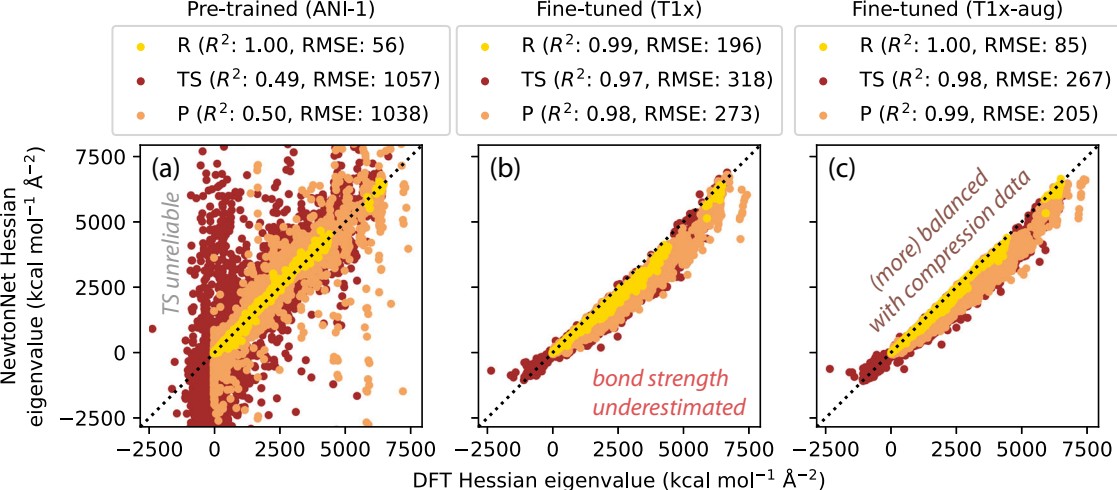

**Fig. 2 | Pre-trained and fine-tuned NewtonNet performance on Hessian prediction of the test set. a** The pre-trained model accurately predicts Hessians at R and P minima geometries but fails dramatically at TSs. **b** The fine-tuned model using the T1x data significantly improves the accuracy at TSs but with notable underestimation of Hessian eigenvalues. **c** Augmenting the T1x dataset with compressed bond configurations creates more balanced training data and improves the overall performance. More comprehensive comparisons of the pre-trained and fine-tuned ML prediction accuracy for Hessians is provided in Supplementary Figs. S4–S8. R: reactant; TS: transition state; P: product; RMSE: root mean squared error; DFT: density functional theory; ML: machine learning. Source data for this figure are provided with this paper.

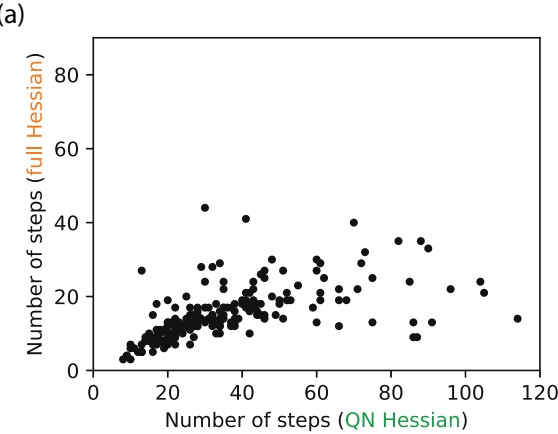

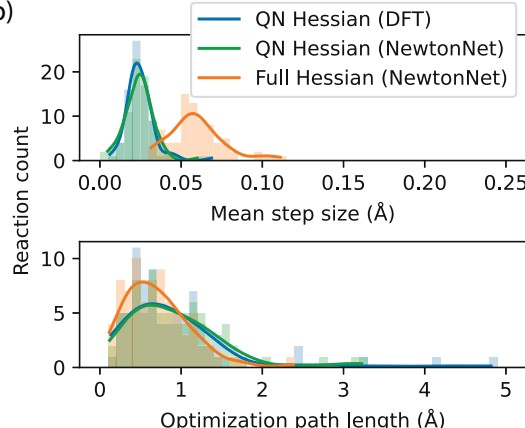

**Fig. 3 | Efficiency improvement using full-Hessian TS optimization compared to the quasi-Newton approach. a** The full-Hessian TS optimization requires 50% fewer steps to reach convergence than the QN approximate-Hessian approach, using the identical NewtonNet potential on the same reactions. **b** The improved efficiency of the full-Hessian TS optimization comes from both more confident steps (top) and more direct paths (bottom) to converge. In this efficiency comparison, gradient calls for initial Hessian construction or Hessian reconstruction for QN restarts have been excluded, whether using DFT or NewtonNet for gradient calculations. TS: transition state; QN: quasi-Newton; DFT: density functional theory. Lines correspond to kernel density estimate fits to the histogram data. Source data for this figure are provided with this paper.

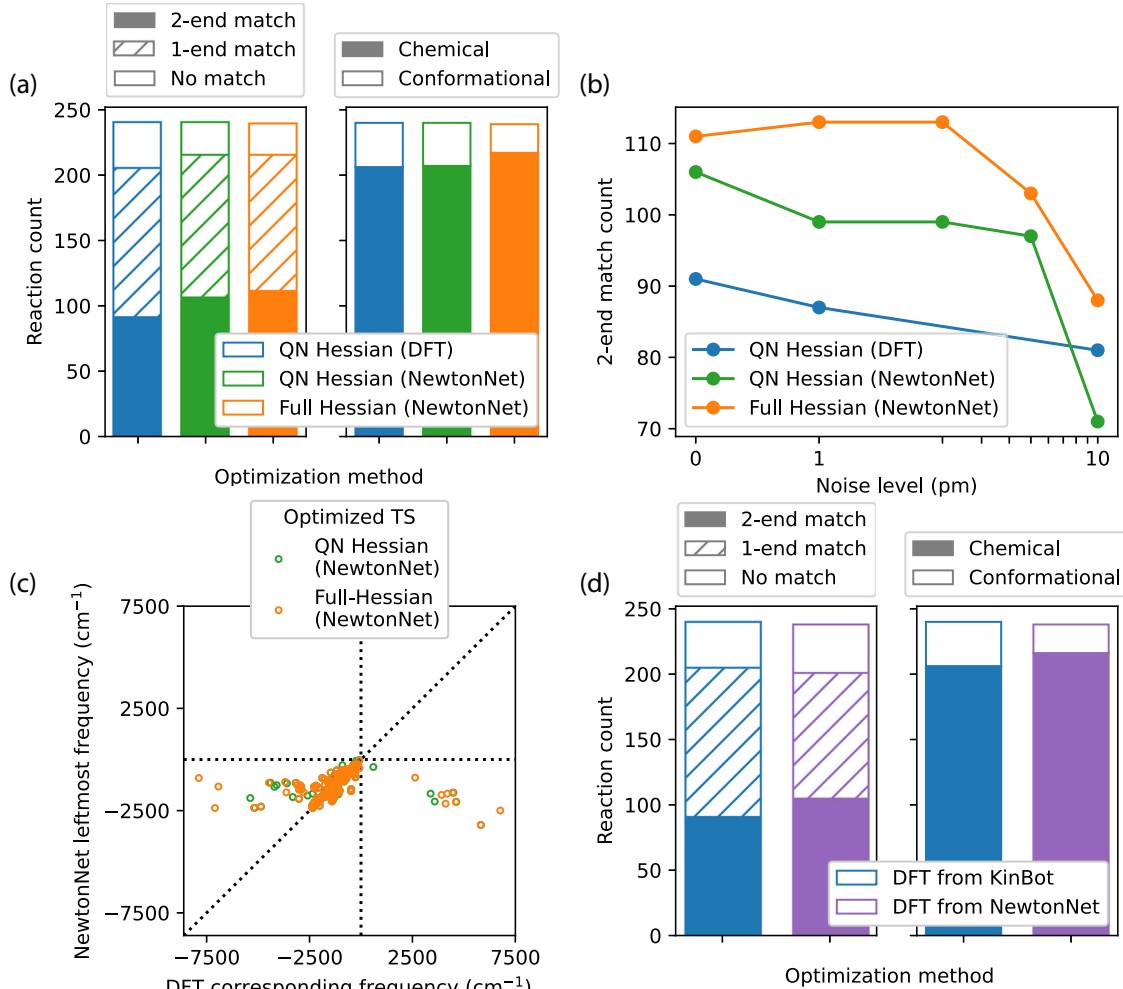

**Fig. 4 | The quality of optimized TSs using NewtonNet. a** NewtonNet predicts reactions that match the intended reactions with higher success rates, both with and without full Hessians, compared to DFT. The full Hessian also finds ~10% more TSs which involve chemical reactions as opposed to conformational changes. **b** The value of the full Hessians over the approximate QN approach is apparent when the quality of the initial TS guesses deteriorates. The QN convergence decays with additional noise to the guess structures, while the full Hessian convergence is more robust to perturbations. **c** Comparing whether the left most frequency found on the ML PES is also a negative frequency on the DFT PES using the ML geometry. **d** Reoptimizing the ML transition state structure on the DFT surface demonstrates superior performance for 2-end matches and identifying chemical reactions compared with starting from the original KinBot initial guess. TS: transition state; QN: quasi-Newton; DFT: density functional theory; ML: machine learning; PES: potential energy surface. Source data for this figure are provided with this paper.

quantitatively illustrates the advantage of using full Hessians over QN Hessians but also shows that their performances can sometimes be qualitatively different. The QN approximation can make the optimization process unnecessarily difficult, even when the underlying problem is not inherently more complex. This observation strongly suggests that the poor convergence in TS optimization is more likely due to the Hessian approximation rather than the quality of the initial guesses or the complexity of the PESs.

Greater optimization efficiency can stem from two factors: increased confidence at each optimization step to increase the step size and a more optimal overall path of optimization. As shown in Fig. 3b, both factors improve the efficiency of the TS optimization with full analytical Hessians. In particular use of the full analytical Hessians exhibits an increase in confidence as measured by the RS-PRFO method, which allows for increased step size on average. A shorter, closer to optimal optimization path also plays a smaller but significant role with the analytical Hessians compared to the QN approach whether on the ML or DFT PES.

Next we consider the robustness of TS optimization using the fully analytical Hessians versus approximate Hessians (using ML or DFT) by

comparing the intended reactions from KinBot to those predicted from the IRC after TS optimization. As shown in Fig. 4a, the NewtonNet full or QN Hessian yielded the intended IRC reactant and product endpoints (2-end match) more often then the QN Hessian from DFT. In some cases only a 1-end match is found because the predicted product is more stable, and more chemically plausible, than the intended product from KinBot on a neutral and closed-shell PES. We also characterize the types of TSs found in Fig. 4a, in which the graphs of the reactants and products are inequivalent for chemical reactions, whereas isomorphic endpoints indicate a conformational TS; in either case, both converge to a first-order saddle point. All KinBot initial guess geometries are intended to yield chemical reaction TSs, and we see that employing full Hessians in TS optimization yields ~10% more TSs that involve chemical reactions as opposed to conformational changes compared to the approximate QN method.

The unreliability of a generated initial guess structure can lead to poor convergence or inaccurate predictions. Therefore, we consider a measure of robustness in which the TS optimization must recover from a poor starting structure, which we analyze by systematically introducing random structural perturbations to the guess structures

generated from KinBot. Figure 4b demonstrates the robustness of NewtonNet's full-Hessian optimization, maintaining consistent performance and even exhibiting slight improvements as noise levels increase up until 2–3 pm. In contrast, the performance using approximated QN Hessians rapidly decays, even using DFT, highlighting the importance of accurate Hessians for robust TS optimization.

A final important metric of robustness is whether NewtonNet predicted TS structures have negative eigenvalues on the DFT PES and/or improve the outcome of TS optimization on the DFT PES. In Fig. 4c we compare the vibrational frequencies of the NewtonNet optimized TS saddle point structures, using both the full Hessian and the QN Hessian, and use those structures as input to calculate the frequencies from the DFT Hessian. We then identify the DFT frequency mode that corresponds to the negative frequency mode from NewtonNet. For ~96% of the reactions, NewtonNet predicts highly accurate imaginary frequencies, regardless of full or QN Hessians. However, there are 10 cases for the QN-ML saddle points, and 11 for the full-ML Hessian saddle points, which have positive DFT Hessian eigenvalues. We further reoptimized all 240 reactions using the full Hessian ML TS structures as initial guesses on the DFT surface (Fig. 4d). In comparison with the optimization outcomes when starting with the original KinBot guess structures, we see an increase in both 2-end matches and chemical transformations under the DFT reoptimization. Thus, starting from ML optimized TS structures we find overall improved solutions on the DFT surface.

## Discussion

We have presented a highly generalizable approach for predicting ab initio Hessians using machine learning based solely on energy and gradient data, and only requiring the property of second-order differentiability. Although Denzel and co-workers found that the feasibility of predicting Hessians using ML without access to explicit Hessian training data was poor[36], our study shows that solely relying on energy and force data using a well-trained ML model can efficiently and accurately predict the Hessian for reactive systems. We attribute our contrasting conclusions to the sufficiently high-quality PES obtained through the deep equivariant message-passing neural network and the mathematical relationship between the potential energy and its derivatives using a ML model that is $C^2$ continuous. To our knowledge, none of the widely used chemical datasets currently include Hessian information, and it is unlikely that such datasets will become available in the foreseeable future due to the high cost of generating ab initio Hessians. Thus, it is good news that models trained on energies and forces are sufficient to derive meaningful ML Hessians.

The ability to generate high-quality explicit Hessians with deep learning models obviates the complexities and assumptions associated with standard TS optimization approaches, in which 240 new reactions never seen in the training data are predicted with greater efficiency, accuracy, and robustness compared to QN ML or DFT. The implementation in Sella to utilize the ML Hessians incurs minimal additional computational overhead and requires no model retraining. Using NewtonNet, the Hessians can be calculated at least three orders of magnitude faster than the DFT calculation, while use of the full ML Hessian takes 2–3 × fewer steps in the TS optimization compared to the QN approach.

This work emphasizes a TS state search methodology and hence used available training data from ANI-1, ANI-1x, and T1x that is specialized for applications involving reactive molecular organic systems but at a low level of DFT and basis set quality. Hence, for accurate predictions it would be desirable to recalculate these data sets at a higher level of theory, likely better density functionals and certainly larger basis sets, in order to predict quantitative barriers. Of course, with appropriate new data sets, we can generalize the ML-Hessian

approach for practical applications of TS optimization in many areas of chemical and material sciences. This is made possible by the tight software integration we have developed for NewtonNet with Sella, and workflows which could be trivially generalized to other relevant ML potentials.

We envision several areas of TS optimization using the ML approach we have described here. For example, the TS discovery for a reaction using methods such as NEB[26,27], QST[28], and GSM[29] often turns to local TS optimization methods when a reasonable approximate structure has been obtained[12]. We showed that our ML transition state structures do improve the DFT optimizations. Therefore, a stepwise procedure that integrates mathematical optimization and machine learning to collaboratively achieve TS discovery for chemical reactions should be viable. In addition, since we can accurately calculate vibrational frequencies at the optimized TS structures, molecular free energy can be efficiently derived using the harmonic approximation. Further, since the quadratic correction can be applied at each step throughout the optimization process, it is possible to optimize TS structures on the free energy surface rather than PES, extending the feasibility of variational transition state theory (VTST) to larger systems where ab initio vibrational analysis becomes impractical[51].

## Methods
### Data preparation
The T1x dataset for training is split in two different ways to assess the accuracy of the model predictions, illustrated in Supplementary Figs. S2 and S3. The original splitting in the literature is based on molecular compositions. All geometries with the same formula (equilibrium and non-equilibrium) are part of the same training, validation, and test set, leading to minimum data leakage. The error from this splitting can be regarded as the worst-case uncertainty estimation of an unseen configuration for the application on the real system in Figs. 1 and 2. On the other hand, we wish to maximize the chemical knowledge from the dataset learned by our model, so a more conventional splitting among molecular conformations is devised to learn the PES of all reactions. Hence in the design of the test set, we ensured that no reaction had both reactant and product pairs found in the training set.

The reference DFT Hessians are performed using Q-Chem 6.0.0[52], using the $\omega$B97X functional[41] and 6–31G* basis set[53] in order to maintain compatibility with the T1x dataset[37]. The eigenvalues are assigned based on the cosine similarity between the predicted and reference eigenvectors as a linear sum assignment problem using the Jonker–Volgenant algorithm[54].

The performance of the ML-Hessian and ML- and DFT-QN TS optimizations is evaluated by the Sella benchmark dataset[39]. The dataset contains 500 small organic molecules between 7 and 25 atoms in configurations that approximate TS geometries across reaction families, among which 265 reactions are closed shell. However, 25 are present in the T1x dataset, thus we subsequently exclude those. We regenerate the remaining 240 such that, in contrast to the original data, the guess structures in our work are constrained minima on an ab initio PES instead of saddles on a semi-empirical PES[37]. We also inject Gaussian noise up to 50 pm directly onto the atomic positions in the Cartesian coordinates of the initial guesses in order to degrade them for the purpose of understanding a given methods ability to still find the TS.

### NewtonNet model and training details
The NewtonNet model with 3 message passing layers is trained using the same architecture as described in Reference[32]. Each node encodes an atomic environment into 128 features initialized by atom types $Z_i$, and each edge encodes an interatomic distance $R_{ij}$ in 20 radial basis functions with a polynomial cutoff of 5 Å[45,55]. The node features are

equivariantly updated with messages from neighboring nodes and edges. The molecular energy $\tilde{E}$ is the sum of all atomic energies $\tilde{E}_i$[56],

$$\tilde{E} = \sum_{i}^{A} \tilde{E}_{i'}(\{Z_i\}, \{R_{ij}\}) \tag{1}$$

where atomic energies $\tilde{E}_i$ are predicted from the node features at the final layer, and $A$ is the total number of atoms. The predicted atomic forces $\tilde{F}_i$ are calculated as the first derivative of the molecular energy $\tilde{E}$ with respect to atomic positions $R_i$[57],

$$\tilde{F}_i = -\nabla_i \tilde{E} = -\frac{\partial \tilde{E}}{\partial R_i} \tag{2}$$

and the predicted atomic Hessians $\tilde{H}_{ij}$ are further calculated as the second analytical derivatives of the energy,

$$\tilde{H}_{ij} = \frac{\partial^2 \tilde{E}}{\partial R_j \partial R_i} = \frac{\partial^2 \tilde{E}}{\partial R_i \partial R_j} \tag{3}$$

However, due to the lack of training data for Hessians $H_{ij}$, only the energy $\tilde{E}$ and forces $\tilde{F}_i$ are trained in the loss function $\mathscr{L}$,

$$\mathscr{L} = \frac{\lambda_E}{M} \sum_{m}^{M} (\tilde{E}_m - E_m)^2 + \frac{\lambda_F}{M} \sum_{m}^{M} \frac{1}{3A_m} \sum_{i}^{A_m} ||\tilde{F}_{mi} - F_{mi}||^2 \tag{4}$$

where $M$ is the total number of molecular graphs, which is 8 million for training, 1 million for validation, and 1 million for testing. After training on energy prediction $\tilde{E}$ and force prediction $\tilde{F}_i$, the model is applied to infer Hessians $\tilde{H}_{ij}$ without further training or fine-tuning.

We use a mini-batch gradient descent algorithm with a batch size of 100 to minimize the loss function using the Adam optimizer[58] with an initial learning rate of $10^{-4}$ and a decay rate of 0.7 on plateau. Fully connected neural networks with sigmoid linear unit (SiLU) nonlinearity[44] for all functions were used throughout the message passing layer. The application of smooth activation functions like SiLU is critical because the network has to be at least twice differentiable for Hessian calculations. We take $\lambda_E = 1$ and $\lambda_F = 20$ in the loss function in Equation (4) to put more emphasis on forces for derivative properties, and an additional L2 regularization of $10^{-5}$ is applied on all trainable parameters to further smooth out the potential energy surface. Layer normalization[59] on the atomic features at every message passing layer is applied for the stability of training. An ensemble of four models is trained on each splitting manner to ensure the reproducibility and reliability of the prediction. An outlier among the 4 predictions is removed if its absolute difference from the closest number compared to the difference from farthest number is larger than the 95% confidence limit of the Dixon Q's test[40].

### Transition state optimization

For the transition state calculations, we use the Quantum Accelerator (QuAcc)[60], a Python package for high-throughput quantum chemistry workflows with an easy-to-use interface for Atomic Simulation Environment (ASE)[61] optimizers. We utilize Sella[39] as the ASE optimizer for TS and intrinsic reaction coordinate (IRC) calculations, having implemented the feature to provide an external Hessian matrix at each optimization step.

Using Sella, the Hessian is automatically transformed into internal coordinates and iteratively diagonalized using the Rayleigh–Ritz procedure for the leftmost eigenpair by a modified Jacobi–Davidson method (JD0, or Olsen's method), with finite difference step size of $10^{-4}$ Å and convergence threshold of 0.1[12]. The TS optimization steps are determined by restricted step partitioned rational function optimization (RS-PRFO)[48–50]. The trust radius is initially 0.1 and adjusted based on the improper ratio (>1) between the predicted and actual

energy change; the radius is increased by a factor of 1.15 when the ratio is below 1.035 and decreased by a factor of 0.65 when the ratio is above 5.0. The IRC is determined by energy minimization at a trust radius of 0.1 Å/amu$^{-1/2}$ in mass-weighted coordinates[62]. QN Hessian updates are achieved using TS-BFGS[22,24]. A maximum of 1000 steps is applied for both TS optimization and IRC search.

The comparison of reactants and products is based on graph isomorphism. Molecular connectivity graphs are created using Open Babel[63] with atom indexing and compared using the VF2 algorithm[64]. The optimization path length is calculated in the Cartesian coordinate distance with the Kabsch algorithm[65]. The path length in Fig. 3b only accounts for reactions with 2-end matches.

### Reporting summary
Further information on research design is available in the Nature Portfolio Reporting Summary linked to this article.

## Data availability
All data[66] including initial transition state guess structures, optimized transition states, and corresponding reactants and products with their coordinates of geometry, energy, forces and Hessians are available at https://doi.org/10.6084/m9.figshare.25356616. Source data for Figs. 1–4 is available with this manuscript. Source data are provided in this paper.

## Code availability
The codebase is comprised of several publicly available packages and tools that contribute to the project. Sella[39] is publicly accessible at https://github.com/zadorlab/sella and comes with comprehensive documentation. NewtonNet[67], another integral part of the project, is also publicly available at https://github.com/THGLab/NewtonNet. The recipes implemented in QuAcc[60] for NewtonNet and Q-Chem, utilizing Sella as the ASE optimizer for transition state and IRC calculations, are publicly accessible and accompanied by thorough documentation at https://github.com/Quantum-Accelerators/quacc. The full workflow and the analysis scripts[68], responsible for generating molecular graphs, retrieving data from the MongoDB database, and performing graph isomorphisms to analyze reactions, are available at https://github.com/THGLab/MLHessian-TSopt. This comprehensive summary provides insights into the availability of the codebase for potential readers and collaborators. Examples to use our end-to-end workflow[69] are available at https://github.com/kumaranu/ts-workflow-examples.

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

## Acknowledgements

S.M.B., A.K., E.C.-Y.Y., and T.H.-G. thank the Lawrence Berkeley National Laboratory Director Research Development program for the work supporting transition state methods. E.C.-Y.Y., X.G., and T.H-G. thank the CPIMS program, Office of Science, Office of Basic Energy Sciences, and Chemical Sciences Division of the U.S. Department of Energy under Contract DE-AC02-05CH11231 for support of the machine learning. E.D.H. and J.Z. acknowledge the Exascale Catalytic Chemistry (ECC) Project supported by the U.S. Department of Energy, Office of Science, Basic Energy Sciences, Chemical Sciences, Geosciences, and Biosciences Division, as part of the Computational Chemistry Sciences Program for work related to Sella. J.Z. acknowledges the U.S. Department of Energy, Office of Science, Office of Basic Energy Sciences, Division of Chemical Sciences, Geosciences, and Biosciences under the Gas Phase Chemical Physics program for work related to KinBot. A.S.R. acknowledges support via a Miller Research Fellowship from the Miller Institute for Basic Research in Science, University of California, Berkeley. This work used computational resources provided by the National Energy Research Scientific Computing Center (NERSC), a U.S. Department of Energy Office of Science User Facility operated under Contract DE-AC02-05CH11231, and the Lawrencium computational cluster resource provided by the IT Division at the Lawrence Berkeley National Laboratory (Supported by the Director, Office of Science, Office of Basic Energy Sciences, of the U.S. Department of Energy under Contract No. DE-AC02-05CH11231).

## Author contributions

S.M.B. and T.H.G. designed the project. E.C.-Y.Y. and X.G. carried out the NewtonNet training and Hessian calculations, E.D.H., A.K. and S.M.B. interfaced NewtonNet with the Sella software package, A.K., S.M.B., and A.S.R. implemented workflows, A.K. and S.M.B. executed TS workflows, J.Z. ran KinBot to generate initial guess structures and reference endpoints, and E.C.-Y.Y. and T.H.G. wrote the paper. All authors discussed the results and made comments and edits to the manuscript.

## Competing interests

The authors declare no competing interests.
