## [Peer Review File · Nature Communications]

Analytical ab initio Hessian from a Deep Learning Potential for Transition State OptimizationREVIEWER COMMENTS

Reviewer #1 (Remarks to the Author):

In this highly innovative and well-written manuscript, the authors learn about initial Hessians using equivariant neural networks. To solve this problem is likely highly valuable to the community, and this, to the best of my knowledge, is among the first attempts to address it.

I recommend accept, despite few minor questions:

- In Equation 4, the Hessians are not incorporated into the loss function. Is there a reason why? It appears that the equivariant NN gets the Hessians right just by interpolating and relying on the smoothness of the function? I'd imagine getting the Hessians from a NN will be trivial compared to the expense of curating datasets.
- 2.1 claims that "The root mean square error (RMSE) of the predictions of eigenvalues is quantitatively quite good given the range of negative to positive values, and the eigenvector similarity between the ML and DFT models also improves significantly after fine-tuning the ML model..." It would be great if the actual RMSE values are highlighted here, as well as in Figure 1 and 4.

Reviewer #2 (Remarks to the Author):

Yuan et al presents a work that utilizes autodiff in machine learning (ML) potential to compute Hessian, which was then used to full-Hessian double ended transition state (TS) search. They show that with full Hessian optimization, the TS search converges in fewer steps. They also compare TS structures optimized by different settings, which show the robustness of their proposed approach despite that predicted Hessian was not 100% accurate. This is the first work, to the best of my knowledge, that uses Hessian in ML potential for TS optimization. Although this is an interesting piece of work, I would like to have some comments and questions been addressed before its publishing.

General comments:

1. The proposed approach would be applicable to all geometric graph neural nets. The authors chose NewtonNet, obviously because some of them developed NewtonNet. However, it would be great to have (at least) one more model for comparison to show the generality of this approach and/or the advantage of NewtonNet.

2. I am not sure that I understand the results regarding to the robustness of ML-enabled full-Hessian optimization (Figure 4a) In the ideal case, should we have all 240 cases as “2-end match”? If that is the case, all three methods are very far away from optimum. But the authors also write “In some cases only a 1-end match is found because the predicted product is more stable, and more chemically plausible, than the intended product from KinBot”. Does this mean “1-end match” is enough? If that is the case, all three methods are quite good, with QN Hessian (NewtonNet) and full Hessian (NewtonNet) performing the same but slightly better than QN Hessian (DFT)? How does QN Hessian (NewtonNet) perform better than QN Hessian (DFT)? The same sets of questions apply to Figure 4d.

3. In double ended TS search, the TS should be unique given the reactants and products. Would it be possible to directly compare the TS structures and their energies optimized by the three methods and the ground truth (basically comparing the black dot and highest point on the red curves in Figure 1c for the reactions and get some statistics)?

4. The author have Transition1x as the training, and use a set of 240 out of distribution reactions as test. Do not get me wrong - it is great to have out-of-distribution test. But would using part of the Transition1x data as in-distribution test be easier to address my questions in point 2 and 3?

Minor comments:

1. I would argue the trend is mostly linear in Figure 3a, at least when QN steps < 80. For cases where the QN steps are larger than 80, the full-Hessian steps are even smaller than those with QN steps < 80, which is difficult for me to understand.

2. The authors show results on reactants and products perturbed by noise. Can we consider a more practical situation, for example, reactants and products optimized by a method cheaper than DFT?

3. In Figure 2, do we have any intuition on why fine-tune TS1x gives worse Hessain predictions on pre-trained (ANI-1) on reactants?

Reviewer #2 (Remarks on code availability):

Given the short period of time for review, I have not gone through the codes. I will start reviewing the code after when the major comments are addressed by the authors.

Reviewer #3 (Remarks to the Author):

The authors have trained a machine learning potential based on an equivariant message passing neural network (NewtonNet) to model gas phase molecular reactions. The training data set contains several million configurations, predominantly from regions around transition states and other parts of reaction pathways and the potential is therefore especially suited for transition state calculations. This is demonstrated by reporting highly accurate energies and forces near transition states. The authors can extract the machine learning potential Hessian matrices by analytic differentiation. The machine learning Hessians have accurate eigenvalue spectrum and very good overlaps of the leftmost eigenvectors with the leftmost eigenvectors of the density functional theory Hessians. The Hessian matrices of the machine learning potential were used for the task of converging guessed saddle point structures to exact saddle points on the machine learning potential energy surface for reactions not in the training data. An algorithm based on restricted step partitioned rational function optimization in internal coordinates (Sella) and recalculating the Hessian at every step was used. The recalculation of Hessians at every step improved the efficiency in terms of iterations by a factor of 2 to 3. A further analysis of the found transition states by following the steepest descent paths and comparing the resulting minima with the reactant and product showed two-end matches in about half of the cases. No overall workflow to find saddle points connecting the input structures with high success rate is shown.

We recognize the novelty of this work in the usage of a training dataset focused on transition states and in the subsequent application of the machine learning potential for transition state search. Machine learning potentials, so far, have mainly been used for structure prediction and molecular dynamics simulation and a broadening of the applicability to transition state search is of great interest. Also, the potential is designed to be twice differentiable and the detailed evaluation of the accuracy of the Hessian of a machine learning potential is an important contribution of the work. The recalculation of the Hessian at every step during the saddle point refinement, even if the number of iterations until convergence to the transition state is significantly reduced, seems less important in our opinion, because this might be a general feature of a smooth potential energy surface, and because the cost of the Hessian calculation might still not be negligible in some cases.

Further comments:

- We are not sure if the title of the manuscript is somewhat misleading, because the Hessian is not directly learned (only the energies and forces appear in the loss function). Though we understand that the terms Hessian and Deep Learning should be in the title, and it might not be possible to avoid this combination.
- There are many cases where the found transition states have no two-end match when following the intrinsic reaction coordinate path (for both machine learning potential and density functional theory) and comparing with the input structures. A further explanation would be helpful. Is it because of an inaccurate initial guess? Or because of intermediate minima involved? Or because of too strict criteria while identifying identical endpoints?
- In Fig. 4(b), the refining of the transition state guess appears to be robust up until a noise level of 2 pm. How is this noise level defined and what unit is it? (A random displacement of 2 pm = 20 Angstrom seems very large.)
- The procedure of creating a saddle point guess with KinBot and refinement with Sella is only a limited use case of machine learning potential trained for transition states. The overall usefulness of the learned potential is not thoroughly proven regarding the lack of a connection between the transition state and the input structures in many cases. It would be of much interest how well other tasks like performing nudged elastic band on the same potential would work.

Reviewer #4 (Remarks to the Author):

I co-reviewed this manuscript with one of the reviewers who provided the listed reports.

This is part of the Nature Communications initiative to facilitate training in peer review and to provide appropriate recognition for Early Career Researchers who co-review manuscripts.

Reviewer 1 (Remarks to the Author): In this highly innovative and well-written manuscript, the authors learn about initial Hessians using equivariant neural networks. To solve this problem is likely highly valuable to the community, and this, to the best of my knowledge, is among the first attempts to address it. I recommend accept, despite few minor questions:

We thank the reviewer for support of our work.

- In Equation 4, the Hessians are not incorporated into the loss function. Is there a reason why? It appears that the equivariant NN gets the Hessians right just by interpolating and relying on the smoothness of the function? I'd imagine getting the Hessians from a NN will be trivial compared to the expense of curating datasets.

As the reviewer accurately pointed out, curating Hessian data is prohibitively expensive compared to energy and force data. Although previous studies have concluded the necessity of incorporating Hessians in training, our work proves that it is possible to infer high-quality Hessians directly from force predictions. Our aim is to leverage the potential of existing deep learning models, which are trained on available datasets without higher derivative properties, to their fullest extent.

- 2.1 claims that "The root mean square error (RMSE) of the predictions of eigenvalues is quantitatively quite good given the range of negative to positive values, and the eigenvector similarity between the ML and DFT models also improves significantly after fine-tuning the ML model..." It would be great if the actual RMSE values are highlighted here, as well as in Figure 1 and 4.

Thank you for pointing this out. The numerical values have been added to the main text.

Reviewer 2 (Remarks to the Author): Yuan et al presents a work that utilizes autodiff in machine learning (ML) potential to compute Hessian, which was then used to full-Hessian double ended transition state (TS) search. They show that with full Hessian optimization, the TS search converges in fewer steps. They also compare TS structures optimized by different settings, which show the robustness of their proposed approach despite that predicted Hessian was not 100% accurate. This is the first work, to the best of my knowledge, that uses Hessian in ML potential for TS optimization.

We thank the reviewer for support of our work.

Although this is an interesting piece of work, I would like to have some comments and questions been addressed before its publishing.

1. The proposed approach would be applicable to all geometric graph neural nets. The authors chose NewtonNet, obviously because some of them developed NewtonNet. However, it would be great to have (at least) one more model for comparison to show the generality of this approach and/or the advantage of NewtonNet.

We used NewtonNet for the important reason that we know that the code works and we implemented Hessians. Nonetheless, as suggested by the reviewer, we have developed a workflow to apply our method with a pre-trained and a fine-tuned MACE model (but using only 10% of the data due to MACE inefficiencies and difficulties) to demonstrate the generality. The workflow and the model, with working examples, are provided in our GitHub repository: <https://github.com/kumaranu/ts-workflow-examples>. We also communicated with the MACE developers and provided assistance with the implementation of Hessians via automatic differentiation in MACE, which we think is beyond scope of our original paper.

However, we would like to make the reviewer aware that we can't realistically do a head-to-head comparison of TS outcomes with the workflow using MACE vs NewtonNet. This would be a completely different paper - a comparison of MLIP architectures would unavoidably distract from the novelty of our results/narrative, and we already showed that one doesn't get one-to-one agreement with DFT and ML

using QN TS optimization. TSs are noisy, and ascribing meaning to specific differences requires careful, time-consuming analysis, as we have shown already for our case-study with DFT. Furthermore, the MACE code seems to have a memory issue that only allows loading of up to ~2M examples, which means we can only train on 20% of the data (NewtonNet does not have this problem). Thus, even if it was desired, an apples-to-apples comparison of MLIPs would require substantial engineering development to make possible, which is also beyond scope of our original paper.

2. I am not sure that I understand the results regarding to the robustness of ML-enabled full-Hessian optimization (Figure 4a) In the ideal case, should we have all 240 cases as ``2-end match"? If that is the case, all three methods are very far away from optimum. But the authors also write ``In some cases only a 1-end match is found because the predicted product is more stable, and more chemically plausible, than the intended product from KinBot". Does this mean ``1-end match" is enough?

Our reaction templates from KinBot are crude for both TS structures and endpoints, which is why we observed many 1-end matches. Some of the endpoints from the templates are not even minima points on the neutral closed-shell PES. Despite this, we did see an improvement in the fraction of 2-end matches when using full-Hessian TS optimization, which can be best seen in Figure 4(b). This improvement is a key point we aim to highlight in our work.

If that is the case, all three methods are quite good, with QN Hessian (NewtonNet) and full Hessian (NewtonNet) performing the same but slightly better than QN Hessian (DFT)? How does QN Hessian (NewtonNet) perform better than QN Hessian (DFT)? The same sets of questions apply to Figure 4d.

The reason why NewtonNet with QN Hessian performed better than DFT with QN Hessian is not entirely clear. While we investigated various hypotheses regarding these differences, such as the smoothness and curvature of the fitted ML PES, we were not able to reach any conclusive findings. This is because TSs are inherently noisy, and it is well known that differences can arise when comparing various electronic structure codes or levels of theory or choice of TS optimization method and we believe this is the best explanation that we can give.

3. In double ended TS search, the TS should be unique given the reactants and products. Would it be possible to directly compare the TS structures and their energies optimized by the three methods and the ground truth (basically comparing the black dot and highest point on the red curves in Figure 1c for the reactions and get some statistics)?

This was our initial approach; however, we encountered ambiguities across different methods (related to the above point). Consequently, we decided to compare only the endpoint graphs, not barrier heights, TS geometries, or other TS-related metrics in this work. Additionally, since DFT with QN Hessians did not always yield 2-end matches, we do not have a clearly established ``ground truth". It is not even clear to us if TSs obtained using DFT with full Hessians (which would be prohibitively expensive) would be any more of a ``ground truth". TS optimization is very sensitive, and there are plenty of examples where QN NewtonNet successfully finds a reaction TS while QN DFT does not, or where QN NewtonNet finds a lower barrier for the same reaction as DFT, or where they find different reactions entirely. When two entirely different reactions are found, is it the relative energies of the transition states that is important, or the barrier heights, or neither? We decided that such a question had no clear answer, and instead established the metrics described in the manuscript.

4. The author have Transition1x as the training, and use a set of 240 out of distribution reactions as test. Do not get me wrong - it is great to have out-of-distribution test. But would using part of the Transition1x data as in-distribution test be easier to address my questions in point 2 and 3?

We appreciate the reviewer's suggestion. Ideally, a model that can perfectly reproduce the ab initio PES would indeed be valuable for benchmarking and diagnosing the method. However, even minor differences

between PESs can significantly impact the TS optimization process. Consequently, it is unclear whether the model, even when tested with in-distribution reactions, can accurately predict the exact forces and infer the precise Hessian matrix.

I would argue the trend is mostly linear in Figure 3a, at least when QN steps < 80. For cases where the QN steps are larger than 80, the full-Hessian steps are even smaller than those with QN steps < 80, which is difficult for me to understand.

As the reviewer pointed out, this behavior not only quantitatively illustrates the advantage of using full Hessians over QN Hessians but also shows that their performance can sometimes be qualitatively different. The QN approximation can make the optimization process unnecessarily difficult, even when the underlying problem is not inherently more complex. This observation strongly suggests that the poor convergence in TS optimization is just as likely due to the Hessian approximation rather than the quality of the initial guesses or the complexity of the PESs. We have added a note to the manuscript to highlight this point. Thank you for the valuable feedback!

The authors show results on reactants and products perturbed by noise. Can we consider a more practical situation, for example, reactants and products optimized by a method cheaper than DFT?

We would like to clarify for the reviewer that we are performing single-ended TS optimization - and thus we are adding noise to the TS guess generated by KinBot, not to reactant or product structures. However, thanks in part to the reviewer's suggestion, we have created a double-ended workflow which allows for more realistic utility. We can now generate the TS guess structure from a geodesic interpolation between a user-provided reactant-product pair, optionally with NEB path optimization, followed by the TS saddle point optimization in this work. Ideally, the TS guess after NEB optimization is of higher quality, and one could compare the TS optimization with and without full Hessians in a more practical setting.

However, we also would like to make the reviewer aware that we can't realistically do a head-to-head comparison of TS outcomes with the workflow using NEB vs Kinbot. This would effectively be a completely different paper, and we already showed that one doesn't get one-to-one agreement with DFT/Quasi-Newton and ML/QN. TSs are noisy, and ascribing meaning to specific differences requires careful, time-consuming analysis, as we have shown already for our case-study. The workflow, with working examples, is available on our GitHub repository: <https://github.com/kumaranu/ts-workflow-examples>, and we will conduct a more comprehensive study in a follow-up paper. }

In Figure 2, do we have any intuition on why fine-tune TS1x gives worse Hessain predictions on pre-trained (ANI-1) on reactants?

As discussed in Section 2.1, we believe this is due to data bias. Reactants are single equilibrium molecules, and since Hessians are derivatives of the atomic forces of the reactants, they fully depend on the model's performance around the equilibrium geometries. A model trained on near-equilibrium data will naturally perform better in this region. When we fine-tune with a TS-rich dataset, we inevitably compromise the model's predictive power at equilibrium points to improve accuracy for reactive species and TSs. While we can recover some accuracy through a proper mixing of datasets, as shown in Figures 2(c) and S8, this process requires balancing the trade-offs.

Given the short period of time for review, I have not gone through the codes. I will start reviewing the code after when the major comments are addressed by the authors.

We also note that we have a new additions to a github in regards changes asked by reviewers.

Reviewer 3 (Remarks to the Author): The authors have trained a machine learning potential based on an equivariant message passing neural network (NewtonNet) to model gas phase molecular reactions. The training data set contains several million configurations, predominantly from regions around transition states and other parts of reaction pathways and the potential is therefore

especially suited for transition state calculations. This is demonstrated by reporting highly accurate energies and forces near transition states. The authors can extract the machine learning potential Hessian matrices by analytic differentiation. The machine learning Hessians have accurate eigenvalue spectrum and very good overlaps of the leftmost eigenvectors with the leftmost eigenvectors of the density functional theory Hessians. The Hessian matrices of the machine learning potential were used for the task of converging guessed saddle point structures to exact saddle points on the machine learning potential energy surface for reactions not in the training data. An algorithm based on restricted step partitioned rational function optimization in internal coordinates (Sella) and recalculating the Hessian at every step was used. The recalculation of Hessians at every step improved the efficiency in terms of iterations by a factor of 2 to 3. A further analysis of the found transition states by following the steepest descent paths and comparing the resulting minima with the reactant and product showed two-end matches in about half of the cases. No overall workflow to find saddle points connecting the input structures with high success rate is shown.

We recognize the novelty of this work in the usage of a training dataset focused on transition states and in the subsequent application of the machine learning potential for transition state search. Machine learning potentials, so far, have mainly been used for structure prediction and molecular dynamics simulation and a broadening of the applicability to transition state search is of great interest. Also, the potential is designed to be twice differentiable and the detailed evaluation of the accuracy of the Hessian of a machine learning potential is an important contribution of the work. The recalculation of the Hessian at every step during the saddle point refinement, even if the number of iterations until convergence to the transition state is significantly reduced, seems less important in our opinion, because this might be a general feature of a smooth potential energy surface, and because the cost of the Hessian calculation might still not be negligible in some cases.

Thank you for the supportive comments. During the revision, we realized that the trend in optimization efficiency not only quantitatively illustrates the advantage of using full Hessians over QN Hessians but also shows that their performances are qualitatively different. We have added a note to the manuscript to highlight this point.

Additionally, the Hessian calculation can be easily vectorized and therefore not much more expensive than a force calculation, at least in gas phase molecular systems that we focus on in this work.

- We are not sure if the title of the manuscript is somewhat misleading, because the Hessian is not directly learned (only the energies and forces appear in the loss function). Though we understand that the terms Hessian and Deep Learning should be in the title, and it might not be possible to avoid this combination.

Thank you for pointing this out. We have revised the title accordingly as ``Analytical ab initio Hessian from a Deep Learning Potential for Transition State Optimization."

- There are many cases where the found transition states have no two-end match when following the intrinsic reaction coordinate path (for both machine learning potential and density functional theory) and comparing with the input structures. A further explanation would be helpful. Is it because of an inaccurate initial guess? Or because of intermediate minima involved? Or because of too strict criteria while identifying identical endpoints?

As the reviewer correctly pointed out, ideally, we would want all 240 cases to result in a 2-end match. However, achieving this is challenging due to the complexity of PESs without high-quality TS guesses. Our reaction templates from KinBot are crude for both TS structures and endpoints, which is why we observed many 1-end matches. Despite this, we did see an improvement in the fraction of 2-end matches when using full-Hessian TS optimization, which can be best seen in Figure 4(b). This improvement is a key point we aim to highlight in our work.

- In Fig. 4(b), the refining of the transition state guess appears to be robust up until a noise level of 2 pm. How is this noise level defined and what unit is it? (A random displacement of 2 pm = 20 Angstrom seems very large.)

In this case, 2 pm = 0.02 Angstrom is actually barely noticeable when plotting the molecular geometry. Even with such a minor perturbation in the geometry, the TS optimization with QN Hessians fails significantly as the Hessians are constructed only approximately. The geometry distortion becomes apparent at around 10 pm, which is where the success rate of all TS optimization starts to decline rapidly. But the main point is that even QN methods degrade even with small perturbations. }

- The procedure of creating a saddle point guess with KinBot and refinement with Sella is only a limited use case of machine learning potential trained for transition states. The overall usefulness of the learned potential is not thoroughly proven regarding the lack of a connection between the transition state and the input structures in many cases. It would be of much interest how well other tasks like performing nudged elastic band on the same potential would work.

Thank you for your suggestion. We have created a double-ended workflow which allows for the TS guess structure to be obtained from a geodesic interpolation between a user-provided reactant-product pair, optionally with NEB path optimization, followed by the TS saddle point optimization in this work.

However, we would also like to make the reviewer aware that we can't realistically do a head-to-head comparison of TS outcomes with the workflow using NEB vs Kinbot. This would effectively be a completely different paper, and we already showed that one doesn't get one-to-one agreement with DFT/Quasi-Newton and ML/QN. TSs are noisy, and ascribing meaning to specific differences requires careful, time-consuming analysis, as we have shown already for our case-study. The workflow, with working examples, is available on our GitHub repository: <https://github.com/kumaranu/ts-workflow-examples>, and we will conduct a more comprehensive study in a follow-up paper. }

Reviewer 4 (Remarks to the Author): I co-reviewed this manuscript with one of the reviewers who provided the listed reports. This is part of the Nature Communications initiative to facilitate training in peer review and to provide appropriate recognition for Early Career Researchers who co-review manuscripts.

Thank you again for very useful feedback

REVIEWERS' COMMENTS

Reviewer #1 (Remarks to the Author):

The revised manuscript has sufficiently addressed all of my concerns. I recommend acceptance now.

Reviewer #2 (Remarks to the Author):

The authors addressed my initial comments and added new experiments. I now recommend it for publication as is.

Reviewer #3 (Remarks to the Author):

We have read the revised manuscript and do not have any objectives against publication. We appreciate that the advantage of the full Hessian approach is now better justified by highlighting the qualitative improvement of the saddle points in some cases. We also understand that a study of the suitability of the potential for other double-ended transition state search purposes would be beyond the scope of the current paper and the authors mention a possible follow-up paper to address this issue.

Reviewer #4 (Remarks to the Author):

I co-reviewed this manuscript with one of the reviewers who provided the listed reports. This is part of the Nature Communications initiative to facilitate training in peer review and to provide appropriate recognition for Early Career Researchers who co-review manuscripts